# Mental Health, Mitochondria, and the Battle of the Sexes

**DOI:** 10.3390/biomedicines9020116

**Published:** 2021-01-26

**Authors:** Paola Bressan, Peter Kramer

**Affiliations:** Dipartimento di Psicologia Generale, University of Padova, Via Venezia 8, 35131 Padova, Italy

**Keywords:** mitochondria, sex differences, sex chromosomes, imprinted genes, diametric theory, kinship theory, parent–offspring conflict

## Abstract

This paper presents a broad perspective on how mental disease relates to the different evolutionary strategies of men and women and to growth, metabolism, and mitochondria—the enslaved bacteria in our cells that enable it all. Several mental disorders strike one sex more than the other; yet what truly matters, regardless of one’s sex, is how much one’s brain is “female” and how much it is “male”. This appears to be the result of an arms race between the parents over how many resources their child ought to extract from the mother, hence whether it should grow a lot or stay small and undemanding. An uneven battle alters the child’s risk of developing not only insulin resistance, diabetes, or cancer, but a mental disease as well. Maternal supremacy increases the odds of a psychosis-spectrum disorder; paternal supremacy, those of an autism-spectrum one. And a particularly lopsided struggle may invite one or the other of a series of syndromes that come in pairs, with diametrically opposite, excessively “male” or “female” characteristics. By providing the means for this tug of war, mitochondria take center stage in steadying or upsetting the precarious balance on which our mental health is built.

Honour thy father and thy mother.*—Exodus 20:12*

## 1. Female versus Male

In mammals, females carry two large, virtually identical sex chromosomes; males, instead, a large and a small one that are quite dissimilar from one another. Hence, in females the effects of a faulty gene sitting on either chromosome can be compensated by a healthy one on the other, but in males it cannot [1]. This may help explain why men suffer more than do women from physical and mental diseases and have a shorter lifespan: across the entire animal kingdom, the sex with the reduced or absent second sex chromosome dies earlier [2].

Even though men are worse off overall, each sex appears especially exposed to problems related to its particular talents (for a review, see [3]). On average, women are better than men in folk psychology [3], that is they possess stronger *mentalistic* skills [4,5]: communicating, maintaining social relations, and reading other people’s facial expressions and nonverbal behavior. They are also better at remembering events (episodic memory), which helps them keep track of who said what, when, where, and to whom [6]. On average, men are better than women in folk physics [3], that is they possess stronger *mechanistic* skills [4]: using tools, identifying objects that are rotated or camouflaged, and navigating through space.

Whether we will develop a man’s or a woman’s brain is normally established quite early during our life as embryos, but there is more to it than carrying a Y chromosome. The Y chromosome is of course critical in the activation of a number of genes that commit the embryo to maleness. The default development plan, which would produce a female, is suppressed and the gonad organizes into a testis rather than an ovary. The testis starts to make testosterone, and it is testosterone that will communicate to the rest of the body, including the brain, that a male should be built (for a fuller, beautifully written account see [7]). That this message is spread around by hormones, rather than by the presence of the Y in each cell, invites all manner of errors and complications. Indeed, the less testosterone boys have as fetuses, the more their later brain development resembles that of girls [8]. Conversely, girls with congenital adrenal hyperplasia, who—already before birth—secrete unusually high levels of testosterone and other androgens, develop in a more male-typical way; showing, for example, better spatial abilities than do other women [9,10]. Even the *adult* brain is rapidly changed toward either male- or female-typical proportions, respectively, by high levels of testosterone and estrogen. These changes are behaviorally relevant, with testosterone addition increasing spatial ability and decreasing verbal fluency, and testosterone suppression doing the converse (as shown in transsexual individuals undergoing hormone treatment [11]). Similar remodelings occur in animals: after testosterone treatment, for example, adult female canaries (that are usually silent) grow song-control brain areas resembling those of males, and start singing like males [12].

The more you have, the more you can lose. As a case in point, Alzheimer’s [13,14] and Parkinson’s [15,16] diseases disrupt language, verbal abilities, and episodic memory in women more than in men; and disrupt visuospatial abilities in men more than in women. The larger impact of Alzheimer’s disease on word recall in women appears related to impaired glucose metabolism: diabetes or insulin resistance—signs of such an impaired metabolism—worsen verbal fluency in women but not in men [17]. Metabolism falls within the purview of mitochondria, former bacteria residing in our cells that supply most of our energy (for a nontechnical review see [18]), and mitochondria cannot do their job if they do not receive enough fuel. Indeed, women who are undernourished due to anorexia nervosa are not as good as other women at using cues from voices, body postures, or facial expressions to read other people’s emotions or intentions—or at putting themselves in somebody else’s shoes (for a meta-analysis, see [19]). At least some of these deficits appear to be direct effects of starvation [19]. Yet recognition of emotions, as measured by the ability to read emotional states from the eyes, does not get any worse in *men* with anorexia nervosa or other eating disorders [20].

## 2. Autism versus Psychosis

The evidence thus suggests that the average woman is better *and* more vulnerable than the average man in tasks that require mentalistic skills, the average man better *and* more vulnerable than the average woman in tasks that require mechanistic ones. Yet most of us are not average but sport a mix of strengths and weaknesses that are not all exclusively mentalistic or mechanistic. In fact, what matters is not whether we are females or males but how much our brain is “female” and how much it is “male”. This brings us to two sets of disorders that may be viewed as pathological extremes of a typically “male” [21,22] and a typically “female” [23,24] brain. The first is the autistic spectrum, which includes autism and related afflictions, such as attention-deficit/hyperactivity disorder and obsessive-compulsive disorder; the second is the psychotic spectrum, comprising schizophrenia and related afflictions, such as major depression and bipolar disorder.

All these syndromes are heterogeneous: patients with the same diagnosis can present vastly different symptoms [25,26]. What the two spectra have in common is a socially debilitating failure to understand other people. In autism, this failure consists in the inability to understand what others might be feeling or thinking; in psychosis, in the delusional overinterpretation of it [4,5]. In fact, autism and psychosis display various characteristics that are diametric opposites of each other ([27]; see also [28,29,30,31,32]) (Figure 1). Mechanistic skills are typically preserved, especially good, or exaggerated in the autistic spectrum, but compromised in the psychotic one. Conversely, mentalistic skills tend to be preserved, especially good, or exaggerated in the psychotic spectrum, but compromised in the autistic one. Susceptibility to these disorders also shows a sex difference, with a male-biased sex ratio in autism [33] and a female-biased one in depression, borderline personality disorder [24], and the hypermentalistic symptoms of schizotypy [34] and schizophrenia [35].

That autism might be male-typical and psychosis female-typical is consistent with the psychiatric repercussions of abnormal dosages of X chromosome. Turner syndrome, in which one of the two X chromosomes is partly or entirely missing, not only reduces the affected women’s risk of psychosis-spectrum illnesses, but it also raises their chances of developing autism or autistic traits [27,36]. Chromosome disorders that feature too much X-chromosome material, such as Triple-X syndrome in women (XXX) or Klinefelter syndrome in men (XXY), produce instead the opposite pattern [37].

The X and Y chromosomes reside in the cell nucleus, but some of the genetic material we carry sits inside the mitochondria. These genes are normally inherited just from the mother, rather than from both parents [38,39]. Since deleterious mutations that harm only males cannot possibly be purged by selection, then, mitochondrial genes have evolved to benefit women more than men [40,41]. Tellingly, mitochondrial DNA variants that reduce male fertility are quite common [42,43]. It has indeed been speculated that mitochondria—which have no evolutionary interest whatever in being in a male—could be selected to kill or feminize males during intrauterine development [44]. Yet this propension would seem only fair, considering the equally provocative proposal that some regions of the Y chromosome may push toward faster implantation and development of male embryos, leading to preferential abortion of females ([45]; for discussion of an alternative mechanism to the same effect, see [46]). The preconditions for a battle of the sexes between mitochondrial and nuclear genes appear to be in place, as the two sets can influence one another in many different ways [47,48].

## 3. Expressing versus Silencing

Both autism-spectrum and psychosis-spectrum disorders are to some extent heritable, but whether and how they will develop also depends on the environment. The environment does not alter the genes themselves, but can change their activity—most notably, whether they are expressed or silenced; this ensures that the organism adapts to the circumstances to the best of its abilities. Such flexibility is obtained through epigenetic processes, which act *upon* (*epi* in Greek) the genes.

As a case in point, these processes control two genes that play a central role in regulating growth: *IGF2*, which during gestation drives the production of the growth-promoting hormone IGF2 (insulin-like growth factor 2), and *H19*, which instead restricts growth. Increased growth appears to go together with the autistic spectrum, decreased growth with the psychotic one [27]. In step with the notion that cancer is a form of overgrowth, the autistic and psychotic spectrum seem associated (albeit not always [49]) with a respectively larger [50] and smaller [51] risk of cancer; children’s cancer risk follows the same pattern as their growth rate [52]. Likewise, variants of genes that favor the growth of nerve dendrites have been associated with autism, those that inhibit it with schizophrenia [53]. Children who were conceived at the peaks of the Dutch and Chinese famines had a sharply higher risk of schizophrenia later in life [54]. It has even been suggested [4] that—via the same mechanism, acting in the opposite direction—developed countries’ overnutrition might be reducing the incidence of psychosis and fuel instead the modern “autism epidemic” [55].

### 3.1. Mitochondria Enable Epigenetics

A particularly prominent epigenetic process involves methylation—the addition of a methyl group to a gene, or nearby protein, so that it can no longer be read. Methyl groups are provided by SAM (S-adenosyl methionine [56]), a remarkably interesting molecule that, after donating its methyl group, is regenerated anew from methionine (or homocysteine, from which methionine can be derived). In humans, methionine cannot be produced from scratch but must be obtained from foods that contain it, such as proteins [57,58], a task that is partly accomplished by mitochondria [48,57]. A study of 60-year-olds conceived during the Dutch famine showed that, compared with their unexposed same-sex siblings, they had less methylation of the *IGF2* gene, hence less silencing of it [59]. This resembles what happens to the offspring of female rats fed a protein-deficient diet starting before pregnancy [60]. The manufacture of SAM requires, besides methionine, the energy-carrying molecule ATP—which again is primarily delivered by mitochondria [18].

Indeed, the symptoms of epigenetic disorders turn out to be strikingly similar to those of mitochondrial disease [57]. After all, to a very important extent, it is the nutrients and energy that mitochondria furnish that enable epigenetics in the first place. And epigenetics does prove to be heavily affected by mitochondria [47,48,61]. For example, the expression of genes that encode the methylation machinery is upset by a common mutation in mitochondrial DNA [62]. Another hint comes from the observations that, first, epigenetic alterations in nuclear genes play a crucial role in cancer; and second, several tumors are accompanied by a depletion of mitochondrial DNA [63]. It so happens that removing the latter from a cell massively changes the methylation of the cell’s genes, and these aberrations can be partly restored by re-introducing mitochondrial DNA back into the cell [63]. Unequivocal evidence of the cross-talk between mitochondria and nucleus has been found in vivo too: hundreds of genes turned out to be abnormally methylated in the offspring of female mice whose mitochondrial function, during pregnancy, had been impaired by rotenone (a natural pesticide) in the diet [64]. This disruption of gene expression in the offspring was permanent and messed up some of the normal epigenetic changes associated with aging—showing one way in which apparently minor events early in life can affect one’s fate much later.

### 3.2. Epigenetic Battles

Conspicuous among the epigenetic processes implicated in both brain growth and brain disease is genomic imprinting—the silencing of either the paternal or maternal copy of a gene before it is passed on to the offspring. At least 228 of our genes are imprinted, an estimate that keeps increasing [65]. This selective silencing of some of the offspring’s genes appears to be in the best interest of one or the other parent: as a matter of fact, men and women find themselves in quite different positions when it comes to parenting. Insofar as only women bear and breastfeed them, children are obviously a larger burden on mothers than on fathers. And of course, within limits, the fewer resources the mother invests in a particular child, the more she will have left for future ones—a problem that is much less pressing for the father, since the mother’s future offspring could easily be conceived with other men.

Unsurprisingly, different circumstances induce different evolutionary strategies. These come to the fore, for example, in how parents go their separate ways in imprinting *IGF2* and *H19*. Tellingly, of the *IGF2* gene, which increases fetal growth, normally the paternally inherited copy is expressed but the maternal one is not. Conversely, of the *H19* gene, which restricts such growth, the maternally inherited copy is expressed but the paternal one is not. It thus appears that with regard to how many resources the offspring should fetch in the early stages of its development, a battle is going on between the parents [66,67].

Indeed, during the time the mother is the only provider, paternal imprinting pushes for more growth-promoting hormones, a bigger placenta to siphon off more maternal resources, more insulin to extract more glucose from the mother’s bloodstream, a larger pancreas with which to produce this insulin, and a robust appetite after birth [68,69]. Maternal imprinting favors instead a less aggressive placenta, and then a smaller baby with little appetite who tends to build up fat reserves, which reduce its need to be fed and kept warm. This parental struggle involves both *IGF2* and *H19*: together, these two imprinted genes are responsible for nearly a third of the variation in birth weight [70]. And there is no denying that the more the fetus grows, the higher the price paid by the mother. To ensure that more nutrients come its way, for example, the fetus can drive up glucose levels in the maternal blood, or raise the latter’s pressure to increase its flow through the placenta, to such an extent that she develops gestational diabetes or gestational hypertension [71]. Complications related to pregnancy or childbirth can endanger the mother’s health and indeed her life (around 300,000 maternal deaths occurred as recently as 2017 [72,73]), while the father risks neither.

After the child has been weaned, when the father is better positioned to lend a hand and possibly become a major provider, paternal imprinting no longer pushes for either growth or appetite. The effects of IGF2 taper off, and the show goes on with other growth hormones that are not affected by imprinting. Maternal imprinting, instead, keeps promoting fat storage and, for the first time, stimulates a big appetite for foods that could just as well be provided by the father or other family members [68,74]. In this way, maternal imprinting encourages fast development and early puberty [67,69], which allows the mother to transfer the bulk of her investment toward younger children or additional ones. Note that, given that his child’s current or future siblings may not be his own, such a priority shift is hardly in the interest of the father.

There might be more to this story than just growth and appetite. It has been suggested ([27,28]; see also [4,5,31,32]) that maternal imprinting would also encourage the development of the offspring’s social and language abilities—folk psychology, mentalistic skills—producing a child that is both easier to handle and better at helping the mother raise younger siblings. As neither trait is especially beneficial to the father, paternal imprinting would instead promote abilities that are of use to the child in dealing with the physical environment—folk physics, mechanistic skills. If maternal and paternal imprinting more or less balanced out, the child would end up with a mix of modest mechanistic and mentalistic strengths and vulnerabilities that would keep it reasonably normal. The more maternal imprinting dominates, the higher the child’s likelihood of developing pathological folk psychology, that is an extreme “female” brain: psychosis-spectrum disorders. The more paternal imprinting dominates, the higher the child’s likelihood of developing pathological folk physics, that is an extreme “male” brain: autism-spectrum disorders (Figure 1).

One typically thinks of the imprinting conflict between parents as a two-way struggle, but there is of course a third party involved. Imprinting involves epigenetic silencing, and for it to proceed normally, the supporting mitochondria must function normally too—which is not always the case.

## 4. (Too Much) Mother versus Father

Because our genes normally come in pairs, one from each of our parents, a defect in one copy can in principle be mitigated or compensated by the other. Of imprinted genes, however, one copy is silenced and thus cannot mitigate or compensate anything. And even if nothing is wrong with the gene itself, a problem can arise with the imprinting mechanism or with the production or dispatching of the nutrients or energy it requires. It is no surprise, then, that imprinted genes are disproportionately often implicated in disease. Interestingly, several imprinting-related disorders also come in pairs: both are caused by an anomaly in the exact same gene, but in one the problem lies with the paternally inherited copy, in the other with the maternally inherited one. And despite the close link between the two disorders, their characteristics tend to be diametric opposites.

### 4.1. Beckwith–Wiedemann Syndrome versus Silver–Russell Syndrome

One such pair of disorders involves the growth-promoting and growth-restricting genes *IGF2* and *H19* on chromosome 11, mentioned before. If *IGF2* is underexpressed or *H19* overexpressed, the baby risks being born with Silver–Russell syndrome; if it is the other way around, with Beckwith–Wiedemann syndrome [58,75,76]. In both cases, the problem can be caused by an alteration in the expressed copy of the gene or in the loss of epigenetic silencing of the normally silenced one [57]. Children with Silver–Russell syndrome (excessive maternal bias) do not grow enough, both in the womb and afterwards (dwarfism), and show little interest in food [77]. Those with Beckwith–Wiedemann syndrome (excessive paternal bias) grow too much both in the womb and afterwards (gigantism) and are especially likely to develop tumors [78,79]. A peculiar feature of this syndrome is a very large tongue—which has been taken to suggest that paternally expressed genes may play a role in the development of the pump that allows the infant to extract milk from the mother’s breast [67]. Also, these children’s odds of presenting autistic symptoms are greater than expected by chance [80].

### 4.2. Angelman Syndrome versus Prader–Willi Syndrome

Another instructive pair of disorders concerns imprinted genes on chromosome 15 [58,75,76,81]. Here an excessive paternal bias can lead to Angelman syndrome, which is characterized by delayed weaning, hyperactivity, sleeplessness, and poor verbal skills—a baby that is a handful for the mother. An overly maternal bias, instead, leads to Prader–Willi syndrome, which typically involves weak suckling, inactivity, sleepiness, and a high pain threshold hence few complaints—a baby that is easy to please. The exaggerated maternal bias that leads to Prader–Willi syndrome showcases a particularly enlightening shift [67]: it pushes birth weight down but adult weight up, and although it reduces suckling in babies, it leaves them insatiable as children and adults—when the mother is no longer the exclusive food provider [74]. People with Angelman syndrome (paternal bias) are especially likely to present autistic-like symptoms [82]; people with Prader–Willi syndrome (maternal bias), psychotic ones [83,84].

### 4.3. Rett Syndrome versus PPM-X Syndrome

One important gene implicated in silencing by methylation, *MECP2*, sits on the X chromosome [85]. The expression of this gene in the cerebral cortex gradually increases until a child is about 10 years old—tracking the progressive maturation of neurons in the period of experience-dependent plasticity. Therefore, the protein regulated by *MECP2* might be repressing genes that are important while neurons are still developing but detrimental afterwards [86]. Mutations of this gene are associated with Rett syndrome and PPM-X syndrome, two disorders that are roughly one another’s mirror image.

Rett syndrome is due to a defective *MECP2* gene on the paternally inherited X chromosome [87] and affects almost exclusively women, as only daughters normally receive this chromosome. The disorder comes with autism-like symptoms [88,89]. PPM-X syndrome is due to a defective *MECP2* gene on the maternally inherited X chromosome; because, in sons, a faulty maternal X chromosome cannot be compensated by a paternal one, men feature the full disorder, whereas women may show no symptoms or a mild intellectual disability only [90,91]. PPM-X syndrome is strongly associated with manic–depressive psychosis [90]. Interestingly, *MECP2* duplication syndrome in boys, in which the *MECP2* gene is not defective but duplicated, is accompanied by autistic symptoms instead [92].

### 4.4. Mitochondria in the Mix

Rett syndrome (which has been studied much more than PPM-X syndrome) has many characteristics in common with mitochondrial disorders, ranging from delayed development and muscle weakness to seizures and heart problems [93,94]. Either directly or indirectly, the *MECP2* gene affects the expression of nuclear genes that regulate mitochondrial functions [94]. And cells of Rett-syndrome patients, just like healthy rodent cells from which the *MECP2* gene has been knocked out, contain malformed, poorly functioning mitochondria that produce little energy and a lot of free radicals [93,94]. Anti-inflammatory dietary supplements that neutralize free radicals, such as fish oil, improve Rett symptoms in girls, suggesting that mitochondrial dysfunction contributes to the syndrome rather than being a mere side effect of it [95]. Furthermore, some patients who, based on their behavioral symptoms, are classified as classic Rett-syndrome patients do not carry a defective *MECP2* gene at all but harbor mutations in their *mitochondrial* DNA [96].

Dysfunctional mitochondria and high levels of free radicals have also been linked to Beckwith–Wiedemann [57], Angelman [97,98], and Prader–Willi [99] syndromes, as well as autism [100,101] and schizophrenia [102,103]. In fact, they have been observed in virtually all mental afflictions [42,104], from chronic psychological stress [105] and fatigue [106] to Alzheimer’s and Parkinson’s disease [107].

Mitochondrial disease can show up with primarily psychiatric symptoms [108], and mutations in mitochondrial DNA are found in patients with all kinds of neurological and psychiatric disorders [109]. Conversely, a single genetic change that lowers the occurrence of mutations in mitochondrial DNA raises the odds that its carriers live to a hundred [110]. All this genetic variant might be doing is reduce by a tiny amount the rate of leakage of free radicals, a nearly imperceptible benefit that may mount up over a lifetime [38]. A convincing argument has indeed been made that, just by slowing down such a leakage, all degenerative diseases linked to aging could be postponed [38]. Note that subtle increases in the occurrence of the most common pathogenic mitochondrial DNA mutation within cells can precipitate abrupt shifts in the expression of nuclear genes—with a lower mutation level manifesting as diabetes or autism, a higher one as multisystem degenerative disease [62]. A high mutation level creates in fact a gene expression profile strikingly similar to that of Alzheimer’s and Parkinson’s diseases [62]. This suggests that the conventional notion that the cause of problems *with* the brain should be searched *in* the brain is naive as well as misleading [111]. It seems likelier that many such problems are driven instead by systemic defects to which the high-maintenance brain is especially sensitive—defects related to the generation and distribution of energy [111].

## 5. Coda: the Inevitable Instability of Mental Health

To best understand a disease, one can zoom in on its details but also zoom out to see how it relates to other ones. The latter is what has been done here: this paper has presented a broad perspective on how mental disorders are tied to the evolutionary strategies of the two sexes and to growth, metabolism, and mitochondria.

The sexes differ in their vulnerability to mental disease, and individuals who sport too much X-chromosome material—too much “femaleness”, so to speak—tend to have mental diseases opposite to those who sport too little. So do, for example, people with Klinefelter or Triple-X syndrome as opposed to those with Turner syndrome. Yet the heart of the matter is not so much being a man or a woman but having, quite irrespective of one’s sex, a female- or male-typical brain. A “female” brain raises the odds of psychosis-spectrum disorders such as major depression, bipolar disease, and schizophrenia; a “male” one, those of autism-spectrum disorders such as attention-deficit/ hyperactivity disorder, obsessive-compulsive disorder, and autism.

Being born with an extreme male or female brain appears to follow from a tug of war between father and mother over how many resources their child should extract from the mother, and thus how vigorously it should feed and how much it ought to grow. An uneven battle not only alters the risk that the child develops insulin resistance, diabetes, or cancer; but can also invite one or the other of a pair of mental disorders with diametrically opposite physical and mental characteristics—genetic syndromes such as Beckwith–Wiedemann and Silver–Russell, Angelman and Prader–Willi, Rett and PPM-X. Mitochondria play an outstanding role in enabling such a battle between our parents (Figure 2), and thus in steadying or upsetting the precarious balance on which our mental health is built.

## Figures and Tables

**Figure 1 biomedicines-09-00116-f001:**
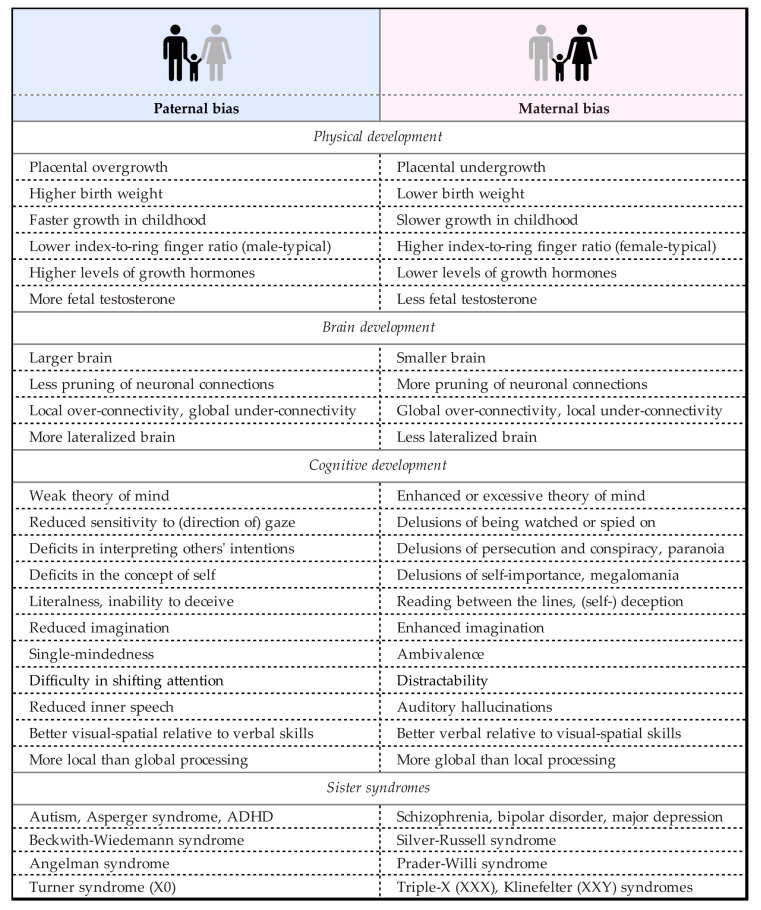
Opposite effects of paternally and maternally biased gene expression on physical, brain, and cognitive development and corresponding disorders of neurodevelopment (sister syndromes). (For references see [27]. Icon: Alina Oleynik from the Noun Project, adapted.)

**Figure 2 biomedicines-09-00116-f002:**
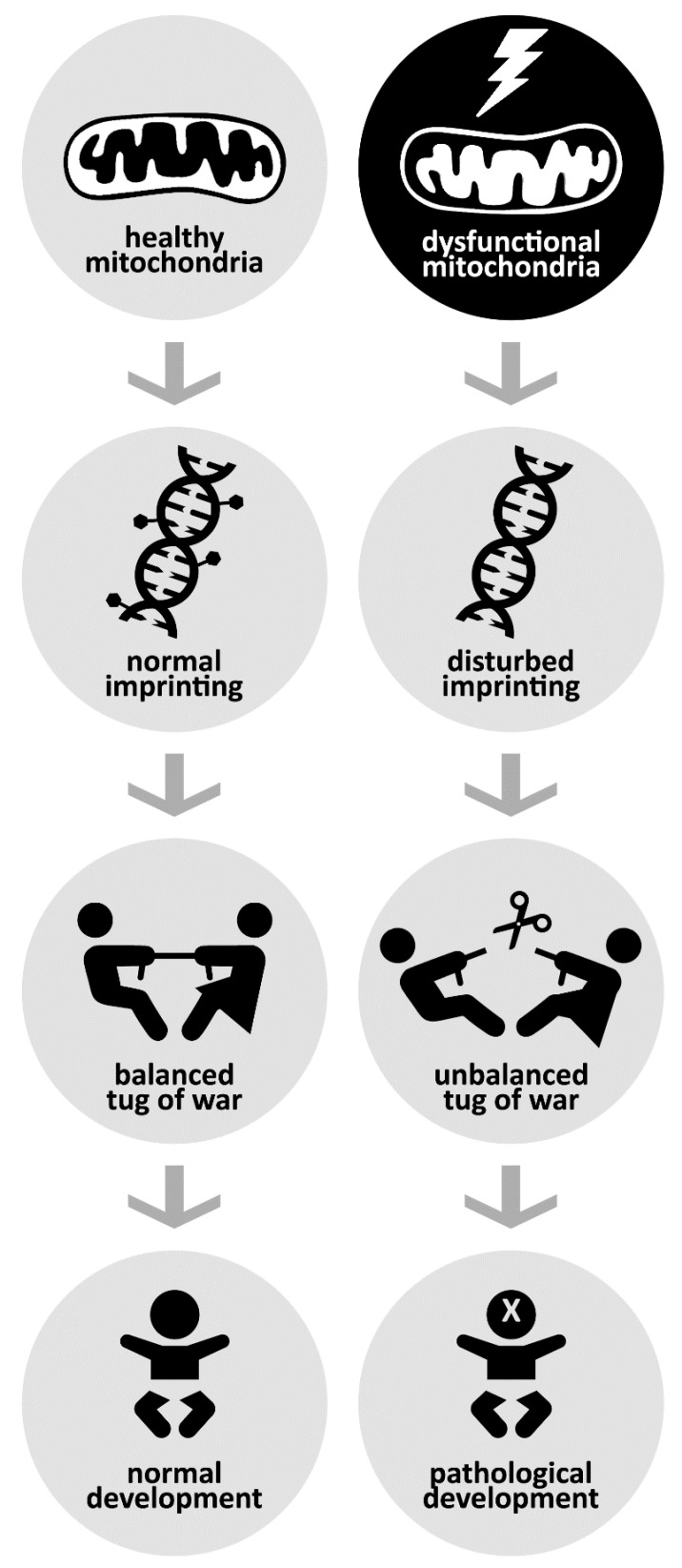
Healthy mitochondria (**left**) enable particular paternal and maternal genes to be silenced (imprinted) with a sort of tag. This results in a roughly balanced tug of war between parents over their genes’ expression in the child—which allows the child to develop roughly normally. Dysfunctional mitochondria (**right**) disrupt imprinting, upsetting the balance and increasing the child’s risk of physical and mental disorders. (Icons adapted from the Noun Project. Lightning: TTHNga; DNA: Léa Lortal; Tug of War: ProSymbols; Scissors, Baby: David.)

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
