# Peer review of "Mental Health, Mitochondria, and the Battle of the Sexes"

_biomedicines, 2021, doi:10.3390/biomedicines9020116_

Round 1
Reviewer 1 Report
Dear Authors
the topic of this review is quite interesting and relevant
given the mental and cognitive problems in oder persons.
Gender appear a risk factors for dementia (see for example: Sex difference in Alzheimer's disease: An updated, balanced and emerging perspective on differing vulnerabilities. Handb Clin Neurol. 2020;175:261-273.) and hormones could mediate these differences (seee for example: 1. The hormonal pathway to cognitive impairment in older men.
J Nutr Health Aging. 2012 Jan;16(1):40-54. 2. Clinical implications of the reduced activity of the GH-IGF-I axis in older men. J Endocrinol Invest. 2005;28(11 Suppl Proceedings):96-100. The paper is well presented and written. Only more results derived by population-based study in older persons are required.
Reviewer 2 Report
In this review article, authors summarized the difference between female and male regarding mental health. This is an interesting review, however, there are several concerns.
If there are no evidences authors can cite, several paragraphs in this article could be considered inappropriate by readers in 2021. (for example, L162-L166 “usually only they wean them with specially prepared transition foods. Children are thus a larger burden on mothers than on fathers, or L186-186 “In the meanwhile, the father has little to lose: if worse comes to worst, he can always look for another mate.”, and so on.
More detailed description about mitochondria and epigenetics/energetic regulation is needed (L132-155). For instance, how is DNA methylation via SAM regulated by mitochondrial function? How can energetic regulation affect epigenome? and so on.
Several mtDNA mutations/deletions are reported to related with mental health such as autism. Please summarize these researches.
Round 2
Reviewer 2 Report
This adequately revised version of review article are interesting and important to the research field. I believe it should be accepted to the journal. Thank you so much for authors' hard work.